# *Trichosporon asahii* Infective Endocarditis of Prosthetic Valve: A Case Report and Literature Review

**DOI:** 10.3390/antibiotics12071181

**Published:** 2023-07-13

**Authors:** Alice Mulè, Francesco Rossini, Alessio Sollima, Angelica Lenzi, Benedetta Fumarola, Silvia Amadasi, Erika Chiari, Silvia Lorenzotti, Barbara Saccani, Evelyn Van Hauwermeiren, Paola Lanza, Alberto Matteelli, Francesco Castelli, Liana Signorini

**Affiliations:** Unit of Infectious and Tropical Diseases, Department of Clinical and Experimental Sciences, University of Brescia and ASST ‘Spedali Civili di Brescia, Brescia 25123, Italy; f.rossini009@unibs.it (F.R.); a.sollima@unibs.it (A.S.); a.lenzi001@unibs.it (A.L.); benni.fumarola@gmail.com (B.F.); s.amadasi@unibs.it (S.A.); e.chiari@infettivibrescia.it (E.C.); silvia.lorenzotti@libero.it (S.L.); barbarasaccani4@gmail.com (B.S.); efu@hotmail.it (E.V.H.); paola.lanza85@gmail.com (P.L.); alberto.matteelli@unibs.it (A.M.); francesco.castelli@unibs.it (F.C.); liana.signorini@asst-spedalicivili.it (L.S.)

**Keywords:** *Trichosporon*, asahii, azoles, endocarditis, prosthetic valve

## Abstract

*Trichosporon* spp. endocarditis is a severe and hard-to-treat infection. Immunosuppressed subjects and carriers of prosthetic valves or intracardiac devices are at risk. This article presents the case of an immunocompetent 74-year-old man affected by endocarditis of the prosthetic aortic valve. After Bentall surgery, cultures of the removed valve demonstrated *Trichosporon ashaii* as the etiological agent. The patient was treated with amphotericin B at first and subsequently with fluconazole. Given the fragility of the patient and the aggressiveness of the pathogen, life-long prophylactic therapy with fluconazole was prescribed. After 5 years follow-up, no drug-related toxicities were reported and the patient never showed any signs of recurrence. The review of the literature illustrates that *Trichosporon* spp. endocarditis may present even many years after heart surgery, and it is often associated with massive valve vegetations, severe embolic complications, and unfavorable outcome. Due to the absence of international guidelines, there is no unanimous therapeutic approach, but amphotericin B and azoles are usually prescribed. Additionally, a prompt surgical intervention seems to be of paramount importance. When dealing with a life-threatening disease, such as mycotic endocarditis of prosthetic valves, it is essential to consider and treat even rare etiological agents such as *Trichosporon* spp.

## 1. Introduction

Invasive mycosis caused by non-*Candida* yeasts such as trichosporonosis are rare opportunistic infections that may lead to life-threatening disease in immunodeficient individuals [1,2,3,4]. Trichosporonosis refers to invasive human infection by *Trichosporon * spp., which is a large heterogeneous genus of basidiomycetous yeasts, producing arthroconidia [5]. *Trichosporon* spp. is ubiquitous in soil, decomposing wood, bird droppings, bats, cattle, food, and water. In rare cases, it can be present among the colonizing flora of the human body, reportedly being isolated from the pharynx, gastrointestinal tract, vagina, skin, and nails [1,6,7,8]. Early descriptions of the pathogen date back to 1890 by Hermann Beigel, who identified *Trichosporon* in infected mustache hair, but several classifications have since been proposed [5]. In 1992, the first taxonomy review was completed, and more species were identified and characterized based on molecular analysis. Several species were recognized as formerly *T. beigelii*: *T. asahii*, *T. mucoides*, *T. inkin*, *T. ovoides*, and *T. steroids* [9,10,11]. Nowadays, the genus includes at least 50 species, 16 of which might have clinical relevance [6,12]. The most common species reported in deep-seated disseminated infections is *Trichosporon asahii*, followed by *T. mucoides*: together, they are responsible for the vast majority of human trichosporonosis cases. *T. ovoides*, *T. inkin*, * T. asteroides*, and *T. cutaneum * are correlated with hair infection and superficial skin lesions [5,13]. The phenotypical identification of *Trichosporon* is based on its ability to produce arthroconidia. However, since *Geotrichum* shares this microbiological feature, when arthroconidia are observed, a urease test is recommended to discriminate *Trichosporon* upon its ability to hydrolyze urea. Phenotypic identification is useful for *Trichosporon* spp. screening, but classical yeast identification methods present limited accuracy and reproducibility. The implementation of new identification methods, such as matrix-assisted laser desorption/ionization–time of flight (MALDI-TOF) analysis, may be useful in the etiological diagnosis. However, only molecular analysis, mostly PCR-based methods, and flow cytometry assays allow the definitive recognition of *Trichosporon* isolates at the species level. The detection of 1,3-β-D-glucan (BDG) is reported in trichosporonosis, but no data are available to define the clinical sensitivity and specificity of this test. Moreover, it is well known that cross-reactions with *C. neoformans* antigens may occur during trichosporonosis [6]. Historically, *Trichosporon* spp. was considered responsible for a distal infection of the hair shaft called white piedra, but was later also associated with invasive infections [1,14,15]. Risk factors for *Trichosporon* spp. invasive infection are neutropenia due to myeloablative chemotherapy, bone marrow or solid organ transplantation, diabetes mellitus, HIV infection, end-stage chronic kidney disease, the positioning of invasive medical devices, extensive burns, prior prolonged antibiotic or glucocorticoid treatment, and previous heart valve surgery. The main pathogenetic hypothesis is that *Trichosporon* spp. penetrates through the intestinal tract, upper respiratory tract, or a venous catheter. This might explain why chemotherapy or prolonged antimicrobial therapy, which are known to be involved in damaging the digestive tract mucosa and causing major disruption of the intestinal flora, respectively, seem to be etiologically related to the onset of trichosporonosis [15,16,17,18]. Invasive *Trichosporon* spp. infections are rarely confined to a single organ. Nonetheless, *Trichosporon* spp. infective endocarditis seems to represent a stand-alone clinical entity, as it is not limited to severely immunocompromised subjects. In fact, the presence of a prosthetic valve or intracardiac device constitutes a major independent risk factor, even many years after cardiac surgery in immunocompetent subjects [15,19]. To the best of our knowledge, no endocarditis on native valves has been reported in the literature in the last 20 years.

We hereby present a intriguing case of *Trichosporon ashaii* endocarditis in an immunocompetent patient who underwent multiple heart surgeries and was later prescribed with life-long medical therapy. Our aim was to highlight the potential aggressiveness of the yeast and the challenges that it may present to the infectious disease specialist.

## 2. Case Report

**Patient history:** We present the case of a 75-year-old Italian male who was referred to our clinic in May 2018. Upon reviewing his medical records, it was discovered that at the age of 39, in 1981, the patient underwent aortic valve replacement surgery. The procedure involved implanting a mechanical valve due to aortic stenosis, suspected to have resulted from post-rheumatic causes. He also had a history of chronic obstructive pulmonary disease (COPD), chronic heart failure, type 2 diabetes mellitus (DM), and obstructive sleep apnea syndrome (OSAS). He reported allergy to amoxicillin. After the surgery and up until 2017, neither post-operative nor long-term complications had occurred. In 2017, a transthoracic two-dimensional echography was performed as a preoperative test before an elective hip replacement surgery. The exam showed a large (5.7) aortic aneurysm and surgical correction of the vascular defect was recommended. The patient underwent a Bentall procedure, consisting of a composite graft replacement of the aortic valve, aortic root, and ascending aorta, with re-implantation of the coronary arteries into the graft. On day 7 after surgery, the patient presented a second-degree atrioventricular Luciani–Wenckebach block, associated with episodes of atrioventricular dissociation. A pacemaker (PM) was therefore implanted. The patient was transferred to the Cardiological Rehabilitation Department. At that time, there was no evidence of leukocytosis (WBC 6.7 × 10^3^/uL, reference range 4 to 10 × 10^3^/uL) or increased inflammatory markers (CRP 2.38 mg/dL, reference range 0 to 0.5 mg/dL). 

**Clinical presentation:** Four months after the cardiac surgery, the man presented to the Emergency Department of our hospital (day 0) with epigastric pain, fever, and diaphoresis. 

**Investigations:** Computed tomography (CT) of the chest, performed to exclude pulmonary embolism, showed two hypodense splenic lesions suggestive of septic embolization. Transesophageal echocardiography (TEE) was performed the day after the admission and revealed a posterior and a medial abscess starting from the valve plane with a 4 cm cranial extension and several floating vegetations attached to the biological aortic valve. Once the suspicion of infective endocarditis was confirmed, a CT scan of the brain and *fundus oculi* examination were prescribed to complete the diagnostic work-up; these excluded cerebral or retinal septic embolization. On day 24, the patient, who was still febrile, was transferred to another hospital. There, he underwent thoracic aortic vascular prosthesis replacement using a bovine pericardial patch from which a tube graft was formed and then sutured with two end-to-end anastomoses. The infected appositions on the prosthetic cups were removed by performing accurate debridement of the previously implanted prosthetic valve. Lastly, a single coronary artery bypass was performed with an inverted autologous great saphenous vein tract to the anterior descending artery. The entire procedure was performed in deep hypothermia. Fifty-two days after hospital admission, the patient was discharged and transferred to a Cardiologic Rehabilitation Institute. One week later, fever started again, and the patient was transferred back to the Cardiothoracic Department of the previous hospital. TEE showed persisting vegetations within the vascular prosthetic lumen and an increase periaortic thickening. Seven weeks after the previous aortic replacement, a new surgical treatment was performed by a second Bentall procedure. Throughout the hospital stay, the PM was never removed, and it is still in place.

**Diagnosis:** From intraoperative samples collected during the last surgical intervention, *Trichosporon asahii* was finally isolated. Identification of the pathogen was made using conventional cultures. Antifungal susceptibility testing demonstrated that the isolated strain was susceptible to fluconazole (MIC 0.250 uL/mL); no other EUCAST breakpoints were established at the time. Periodically collected blood cultures consistently showed no growth. 

**Treatment:** Empirical treatment with vancomycin, gentamycin, and rifampin for prosthetic valve endocarditis was started three days after hospital admission. On hospital day 17, the patient was still febrile, so gentamycin was suspended, and levofloxacin and cefepime were introduced. One week later, considering the lack of clinical improvement, the therapy was modified again: vancomycin, levofloxacin, and cefepime were suspended and meropenem and daptomycin, along with ongoing rifampin, were started. After the first surgical intervention, all ongoing therapy was suspended and switched to daptomycin, ceftriaxone, and levofloxacin. Then, before the second surgical intervention, treatment was modified again to levofloxacin, meropenem, daptomycin, and anidulafungin. As soon as *Trichosporon asahii* infection was diagnosed, intravenous amphotericin B was introduced, and the ongoing antibiotic therapy was suspended a few days later. Ten days later, the antimicrobial treatment was simplified to fluconazole (600 mg every 24 h). The patient was discharged seven weeks after the first hospital admission. 

**Follow-up:** During the scheduled follow-up, he underwent imaging controls with multiple FDG-PET/CT (18F-fluorodeoxyglucose positron emission tomography, contrast-enhanced computed tomography), which repeatedly showed conspicuous hyperaccumulations in the ascending aorta up to the aortic valve. Blood parameters were monitored every two weeks and no significant changes were found in the first eight months: blood count was stationary; C-reactive protein (CRP) was always less than 10 mg/dL (reference range 0 to 5 mg per liter); aspartate aminotransferase and alanine aminotransferase results were in range in all measurements; and the blood level of creatinine was stable. Fluconazole was well tolerated and discontinued after 12 months. During the months following the end of the therapy, FDG-PET/CT remained unchanged. Due to anamnestic COPD, the patient required several prolonged steroid treatments because of frequent flare-ups. CT scan of the chest showed lung densities in the middle lobe surrounded by ground glass areas. Blood tests showed increased CRP (29 mg/L in a first episode and 19.6 mg/L in the second event; reference range, 0 to 5 mg/L). Considering the higher risk of recurrence of complicated fungal infection in immunocompromised subjects, the patient’s frailty, and his frequent need for steroid treatment, we decided to resume antifungal therapy, and fluconazole 100 mg daily was prescribed as life-long suppressive therapy. Approximately one year after the reintroduction of therapy, increased bilirubin (1.75 mg/dL; reference range, 0.2 to 1.2 mg per deciliter) and gamma-glutamyl transpeptidase (195 U/L; reference range, 0 to 50 U/L) were noted. On suspicion of liver toxicity, we decided to halve the dosage of fluconazole (50 mg every 24 h). In the following years, the patient went through a progressive normalization of cholestasis indices. The semiannual TTE follow-up showed substantial stability. Ejection fraction was 30%. Mitral valve revealed fibro-calcification. Doppler study revealed mild-to-moderate mitral regurgitation. Aortic biologic prosthesis showed normal opening, no obstruction, no evident pathological regurgitation, and no significant stenosis. At the last outpatient evaluation, five years after the start of suppressive therapy, the patient was found in good general condition and reported subjective well-being. On physical examination, vesicular breath sounds were reduced in left pulmonary base with no added pathological sounds. The cardiac examination was normal, except for a grade 3 systolic murmur. The remainder of the examination was normal. A graphic representation of the timeline of the clinical course of the patient is shown in Figure 1. 

## 3. Literature Review

Advanced research on PubMed for “(trichosporon) AND (endocarditis)”, filtering the last twenty years (from January 2003 to April 2023), generated eight results, all of which were case reports. One of them was inherent to *Wickerhamomyces anomalous* bacteriemia [20]; therefore, we only considered seven research papers in our analysis, in addition to our case report. The main clinical information concerning the papers is summarized in Table 1. 

Our search showed that trichosporonosis patients are mostly men (only one woman was reported with the disease), with a median age of 56.00 ± 16.16 years. *Trichosporon* endocarditis developed after a median of 5.60 ± 4.56 years from heart surgery. Surprisingly, even if *Trichosporon* spp. is a known opportunistic pathogen, all patients were immunocompetent. A possible explanation is that these cases are more likely to be published on account of their singularity. Alternatively, it may be speculated that previous heart surgery is such a strong risk factor that *Trichosporon* endocarditis can occur in immunocompetent subjects. Clinical presentation was usually subacute with low-grade fever, leukocytosis, and CRP elevation. Eosinophilia was reported by two authors. A previously unknown cardiac murmur was described in 4/8 cases; microvascular damage signs, such as conjunctival hemorrhage, petechial eruption, or Osler nodules were present in 4/8 patients; and hemiplegia was described in 1 case. Interestingly, a correlation to onychomycosis from *Trichosporon mucoides* is pointed out in one case: this was the only fungal-related clinical sign at time of hospital admittance. All the patients had a previous heart surgery, mostly because of valvopathies, and one PM implant. In all cases, the left chambers of the heart were implicated, with the only exception of the PM infection, which was positioned in the right chambers. Four cases of endocarditis involved the aortic valve (in one case also the wall of the ascending aorta presented a vegetation), one involved the mitral valve, and in two cases, vegetations on both the left valves were documented. The most impressive feature of these vegetations was their very large size (up to 2.54 × 2.36 cm). This is probably the reason why most of the echocardiographic diagnoses were made by a transthoracic echocardiography (TTE in 4 cases, TEE in 3 cases, unspecified in 1 case). The large size of vegetations, often described as mobile and irregular, may also explain the high incidence of severe complications reported. In detail, embolic complications, both splanchnic and encephalic were demonstrated in 5/8 patients, and hemiplegia/hemiparesis were described in 2 cases, vasculitis-like complications were demonstrated in 2/8 cases, and heart failure occurred in 3/8 of the patients, with a rapid fatal evolution in 2 of them. 

*Trichosporon* spp. was found in blood cultures in 5/8 patients, and in valve cultures in the 3/8 cases with negative blood cultures. It was the only proven etiologic agent in all but one case, in whom a *T. asahii* fungemia complicated a previously documented endocarditis of prosthetic valve from *S. oralis*. In one case, the cultural diagnosis was confirmed by a histopathological exam. An antigenic test was performed in two patients, both with positive results (BDG and galactomannan in one case and BDG only in the other one). 

Patients presented with signs of endocarditis, but no specific risk factor or clinical sign suggestive for a mycotic etiology; therefore, most patients initially received empirical antibacterial therapies (most frequently the combination of vancomycin, gentamycin, and rifampin). In one single case, caspofungin was initiated for suspected candidemia.

Different approaches to treatment of trichosporonosis were adopted due to the lack of international guidelines and the very large time window considered. Liposomal amphotericin B was prescribed as monotherapy in two cases, and in 3/8 cases in combination with azoles (1 ketoconazole, 2 voriconazole). In additionally, 2 patients were treated with azole monotherapy (1 fluconazole, 1 voriconazole). After a variable amount of time, they were all switched to a maintenance therapy with azoles. Five patients underwent a surgical source control of infection, while life-long voriconazole suppressive therapy was prescribed to the other three. The difference between these two treatment strategies was evaluated based on the clinical outcome. Both groups presented a fatality, while the others had a clinical resolution with no recurrences. However, 2/3 patients in the suppressive therapy group presented increased size of vegetations at follow-up echocardiograms.

## 4. Discussion

The etiologic agents most frequently isolated in mycotic endocarditis are *Candida* spp. and *Aspergillus* spp. [24], but there is increasing clinical interest in rare yeast etiologies. The incidence of trichosporonosis in deep mycosis patients is about 5% [17]. Trichosporonosis is commonly found in immunocompromised patients, and the associated outcome is usually poorer than other similar yeast-like invasive infections, as mortality ranges from 30 to 90% [1,6]. Emboli complications have been reported to occur in 75% of the cases, and mortality has been reported up to 83% [15,16]. In some cases, the *exitus* may occur before starting treatment or even before diagnosis [21]. Known high-risk patients for *Trichosporon* spp. deep-seated infections include those who underwent bone marrow or solid organ transplantation, burn victims, receivers of prosthetic heart valves or intracardiac device placement, HIV-infected subjects, those undergoing peritoneal dialysis, and intravenous drug users [16,25]. Some authors reported the genus *Trichosporon* to be the second most common agent of disseminated yeast infections after the genus *Candida*, in patients affected by a malignant hematopathy [6,26,27,28]. In these patients, the outcome is usually worse. However, there are some exceptions; for example, invasive infections due to *T. mucoides*, including fungemia, peritonitis, brain abscess, and prosthetic valve endocarditis, have been reported in both immunosuppressed and immunocompetent hosts [1,15,29,30,31,32]. The diagnosis of trichosporonosis includes various tools that can often be used in combination. When *Trichosporon* mycelial threads or conidia are observed in a biopsy specimen, they must be considered as definitely diagnostic. However, it is not advisable to rely on a histopathological diagnosis because adequate samples might not be available, and it usually takes a long time to get the results. Therefore, blood cultures, or any culture of otherwise sterile samples (such as cerebrospinal fluid, endothelial tissues) are the only established microbiologic diagnostic tool [1,6]. However, blood cultures have low sensitivity, resulting in false-negative results in systemic trichosporonosis [14,16,33]. Whereas the diagnostic role of histopathological exam, or positive blood cultures is well defined, the usefulness of antigenic tests is not. For example, both galactomannan and BDG elevation could be correlated to the presence of *Trichosporon* spp. However, since their specificity is very low, they could play a better role as screening markers for mycotic infections in general, rather than trichosporonosis [1,16]. Moreover, a cross-reactivity with glucuronoxylomannan antigen, which is known as the *Cryptococcus* capsular antigen, has been described [34]. As an indirect diagnostic tool, eosinophilia may be associated with trichosporonosis. In rare cases, eosinophils may reach even very high values and even lead to acute eosinophilic pneumonia. When an elevated eosinophil count is reported at diagnosis, it can be later used as a marker of response to treatment [23]. The clinical presentation of *Trichosporon* spp. endocarditis is comparable to any other infective endocarditis. It is recommended to follow the revised Duke’s criteria to risk stratify patients into three categories of definite, possible, and rejected diagnosis [35]. Regarding our case, it was correctly considered as a definite infective endocarditis according to Duke’s criteria. The physiopathology of *Trichosporon* spp. endocarditis seems to be strictly dependent on the presence of a prosthetic valve or intracardiac implant [1,6]. In some cases, the pathogen reaches the heart valves after contamination of a percutaneously inserted intravascular catheter via colonized skin [36]. This is probably how our patient acquired *Trichosporon asahii*, right after the heart surgery. However, the trichosporonosis clinical course is peculiar, and the infection timing may also be prolonged as *Trichosporon* spp. is not necessarily a hospital-acquired pathogen. As our literature research underlines, *Trichosporon* spp. endocarditis may present even after eleven years from the implant of a prosthetic valve [14,15,16,19,21,22,23]. An underlying gastrointestinal colonization and further translocation throughout the gut seems to be the most probable explanation to this kind of late-onset deep-seated infection [37,38]. 

The optimal treatment of trichosporonosis is not established yet due to the rarity of the infection. According to the global guideline for the diagnosis and management of rare yeast infections, posaconazole or voriconazole are considered as first-line treatments, and fluconazole may be acceptable as an alternative first choice. Amphotericin B is only contemplated as a second-line treatment. Echinocandins should be avoided [1] as all *Trichosporon* genera had been found to be intrinsically resistant to these drugs in vitro [6,39]. In our case, amphotericin B was chosen for induction therapy and a good outcome was obtained, as in other case reports [19,23]. A combination therapy with 5-fluorocytosine and amphotericin B has also been associated with good clinical response in one case report [29]. Moreover, combination therapy with 5-fluorocytosine and amphotericin B has been associated with good clinical response in one case report [29]. Nevertheless, in other studies, amphotericin B has been correlated to poor clinical outcome, and therefore it may not be a good option as a monotherapy [40,41]. Azoles, and especially the newer triazoles, may be a good therapeutic strategy both as monotherapy and combination therapy. Fluconazole, posaconazole, voriconazole, and ravuconazole have been described to be efficacious; in particular, voriconazole presented the strongest activity in vitro, even better than liposomal amphotericin B [1,14,15,30,42,43]. The use of terbinafine, which is usually involved in the treatment of dermatophyte infections, has been reported in one case report in association with voriconazole [23]. The rationale of these association is that in vitro a synergistic combination has been reported with azoles, echinocandins, or polyenes. However, evidence suggests an emerging resistance to azole treatment because of the formation of biofilms; this hypothesis is particularly concerning for patients with cardiac devices or mechanical prosthetic valves [44]. Therapeutic failure has been described, mostly in immunocompromised patients, for all cited antimycotic regimens.

In general, when fungal endocarditis is diagnosed, pharmacological therapy is not sufficient: a valve replacement should be considered mandatory [15,23,35]. The 2009 IDSA guidelines for Candida endocarditis treatment recommend fluconazole as step-down therapy for patients undergoing valve replacement surgery, and indefinite fluconazole suppression therapy for patients unable to undergo valve replacement surgery [45]. The American Heart Association guidelines recommend valve surgery in addition to parenteral antifungal therapy, followed by life-long therapy with azoles [35]. Boland et al. [46] described their 40-year experience (1970–2008) at the Mayo Clinic with fungal endocarditis. They collected twenty-one cases of culture-proven prosthetic valve fungal endocarditis. Among the seven long-term survivors (follow-up range, 22–129 months; mean, 70 months; median, 72 months), four received chronic (>6 months) suppressive treatment after initial treatment with valve replacement and parenteral antifungal therapy. Given the limited experience on *Trichosporon* spp. endocarditis, there are no official recommendations regarding long-term suppressive therapy and, nowadays, the prescription of long-life suppressive therapy after the valve surgical replacement is controversial. In fact, considering the ability of *Trichosporon* spp. strains to form biofilms on implanted devices, azoles or even the newer triazoles might not represent the optimal molecules for suppressive therapy after cardiac surgery because of their poor penetration of biofilms [44,47,48,49]. However, since an endogenous dissemination model of *Trichosporon* spp. have been theorized, at least in immunocompromised or at-risk patients, this therapeutic approach may be useful in preventing recurrencies of trichosporonosis in previously affected patients [6,50]. In our case, the fluconazole prescription was always well-tolerated and possibly contributed to our patient remaining relapse-free.

## 5. Conclusions

Clinicians should be aware of the existence of rare causes of fungal endocarditis or endovascular infections such as *Trichosporon* spp. in immunocompromised and immunocompetent patients with recent heart surgery. Consideration of all possible causative agents of infective endocarditis is essential to prescribe an appropriate empirical therapy. Considering the high mortality rate of *Trichosporon* spp. invasive infections, this yeast should be considered when treating a prosthetic valve infection with negative blood cultures, irrespective of the time elapsed since surgery. The combination of voriconazole plus amphotericin B, along with prompt surgical intervention, may be effective for the acute treatment of *Trichosporon* endocarditis. Considering the high mortality and morbidity risk in case of fungal endocarditis recurrencies, long-term suppressive therapy with azoles should be considered in patients who have undergone surgical treatment and are considered at high risk of infection relapse.

## Figures and Tables

**Figure 1 antibiotics-12-01181-f001:**
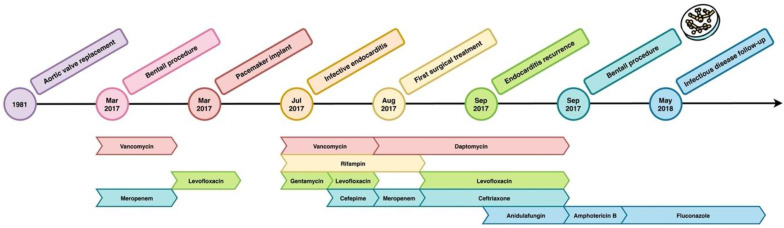
Timeline of the clinical history of the patient and subsequent antibiotic molecule administrated. The little Petri dish indicates the isolation of *T. asahii* on valve samples.

**Table 1 antibiotics-12-01181-t001:** Summary of cases of Trichosporon asahii endocarditis from January 2003 to April 2023. The present case is included (number 8). M: male, F: female, IV: intravenous, HTN: hypertension, FU: follow-up, mth: month, NA: not applicable.

	Age, Sex, Country	Comorbidities and Previous Surgery	Cultures and Pathogen	Site of Infection	Induction Therapy	Maintenance Therapy	Cardiac Surgery	Medical Suppression	Clinical Outcome	Follow-Up
1 [21]	52, M, Spain	Biological valve replacement	Blood cultures: *T. inkin* post-mortem	Vegetation of the prosthetic aortic valve	None	None	None	None	Exitus 30 h after hospital admission	NA
2 [16]	58, M, Japan	HTN	Blood cultures: *S. oralis* later *T. asahii*	Endocarditis of prosthetic mitral valve and aortic valve cusp	IV fluconazole 400 mg/day	Fluconazole 200 mg q24h	Aortic and mitral valve replacement	None	Recovery	NA
3 [14]	57, F, India	HTN, DM, COPD, PM implant	Blood cultures: *Trichosporon* spp.	PM infection all valves were normal	IV voriconazole 200 mg q12h	Oral voriconazole 200 mg q12h	None	Voriconazole 200 mg q12h	Recovery	No recurrence, increased size of vegetation
4 [22]	20, M, Texas, USA	Biological aortic valve replacement	Blood cultures: negative Valve cultures: *T. asahii*	Large vegetations on the anterior wall of the aorta	Amphotericin B and ketoconazole for 2 weeks	Voriconazole	Valve replacement, patching of the aorta’s wall	Voriconazole 400 mg q8h for 6 months, q12h for 6 months, qd for 6 months	Complete recovery, resolution of hemiplegia	Normal echocardiogram at 5 years FU
5 [15]	66, M, Republic of Korea	Aortic valve replacement and mitral annuloplasty	Blood cultures and valve cultures: *T. mucoides*	Endocarditis of the prosthetic aortic and mitral valves	IV voriconazole and amphotericin B for 22 days	Voriconazole for 3 months	Valve replacement, removal of the prosthetic ring and pannus	None	Recovery	No evidence of relapse at 18 months FU
6 [19]	57, M, Portugal	Mitral valve replacement	Blood cultures: negative Valve cultures: *T. beigelii*	Endocarditis of prosthetic biological valve	Amphotericin B	Voriconazole	Valve replacement	None	Exitus 13 days after surgery	NA
7 [23]	63, M, Canada	DM, bioprosthetic aortic root and valve	Blood cultures: *T. mucoides*	Endocarditis of prosthetic aortic valve and valve abscess	Amphotericin B q24h for 6 days, IV voriconazole q12h for 2 weeks	Voriconazole 300 mg q12h and terbinafine 250 mg q24h	None	Voriconazole	Recovery	No recurrence but increased valve abscess at 12 months FU
8	75, M, Italy	COPD, DM, OSAS Bioprosthetic aortic root and valve	Blood cultures: negative Valve cultures: *T. asahii*	Abscess of bioprosthetic aortic valve and several vegetations	Amphotericin B for 10 days	Fluconazole 600 mg q24h	Bioprosthetic valve replacement	Fluconazole 400 mg, later reduced to 100 mg, later reduced to 50 mg	Complete recovery	No recurrences, no valve vegetations at 5 years FU

## Data Availability

No new data were created or analyzed in this study. Data sharing is not applicable to this article.

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
