# Peer review of "Trichosporon asahii Infective Endocarditis of Prosthetic Valve: A Case Report and Literature Review"

_antibiotics, 2023, doi:10.3390/antibiotics12071181_

Round 1

Reviewer 1 Report

The authors herein present a case report and review of Trichosporon spp infective endocarditis of prosthetic valves. The manuscript is quite extensive and provides a robust analysis on the subject. The authors should also be congratulated for symmarizing analogous reports that were published in the last two decades. I have the following comments

1. The procedure that was performed in lines 112-114 is poorly described. Did you perform an Ozaki aortic valve neocuspidization using xenologous pericardium?

2. Line 116: "..the anterior descending branch had gone in deep hypothermia.." What does that even mean?

3. Can you add pertinent imaging from your own case (TEE, intraop pictures etc)? 

4. Any thoughts on the virulence of the pathogen in immunocompromised patients?

5. Minor - you need to improve the structure of the text by at least diving it in more paragraphs 

Please have this reviewed by a native English speaker.

Author Response

Dear reviewer,

I would like to thank you very much for your kind revision on behalf of all the authors.

We appreciate your contribution and your evaluation of our work.

  1. Concerning lines 112-114, the performed procedure was not an Ozaki aortic valve neocuspidization using xenologous pericardium. Thanks to your kind suggestion, we elaborated the surgical procedure as follows: “Here, he underwent thoracic aortic vascular prosthesis replacement using a bovine pericardial patch from which a tube graft was formed and then sutured with two end-to-end anastomoses. The infected appositions on the prosthetic cups were removed performing an accurate toilet of the previously implanted prosthetic valve. Lastly, a single coronary artery bypass was performed with an inverted autologous great saphenous vein tract to the anterior descending artery. All the procedure was performed in deep hypothermia.
  2. The entire surgical intervention was performed in deep hypothermia, as meaning that the temperature was set at 20°C.
  3. Unfortunately, no pertinent imaging from our own case report is available anymore due to a renewal of the informatic system in the Hospital where our patient was operated.
  4. The virulence of the pathogen in immunocompromised patients is extensively described in literature, as we tried to sum up in the introduction paragraph. T. asahii is usually more virulent in these hosts, and it can cause severe deep-seated infections. However, endocarditis of prosthetic valves is mostly described in immunocompetent patients (such as in our case report). Therefore, all the subjects who underwent a heart surgery should be considered at risk, especially in case of a prosthetic valve replacement or pacemaker implantation.
  5. Thanks to your kind suggestion we divided the text of the case report and the discussion in more paragraphs.

A native English speaker has now read and revised our manuscript, we incorporated all his amendments to the new draft of the manuscript. We truly hope you will find this new version of the manuscript to be more organized and understandable.

Thank you for your help, we hope that you will read and appreciate our final paper.

Reviewer 2 Report

1.     Please rephrase lines 78-81 into : "We present the case of a 75-year-old Italian male who was referred to our clinicin May 2018. Upon reviewing his medical records, it was discovered that at the age of 39, back in 1981, the patient underwent aortic valve replacement surgery. The procedure involved implanting a mechanical valve due to aortic stenosis, suspected to have resulted from post-rheumatic causes.

2.     It’s very difficult to follow the case report part, I suggest restructuring into, or other  subchapters, as you wish.

Clinical presentation

Investigation

Diagnosis

Treatment

3.     The literature review part should be presented before the case report section.

4.     The patient was moved to the Cardiological Rehabilitation Department right after discharge. ….rephrase into “ The patient was transferred or was addressed”

5.     Their manuscript should undergo extensive English editing.

6.     No images from CT or echocardiography. If available, please add.

7.     Overall, the case is interesting, but the presentation form required major modifications.

extensive editing

Author Response

Dear reviewer,

I would like to thank you very much for your kind revision on behalf of all the authors.

We appreciate your contribution and your evaluation of our work.

  1. We thank the reviewer for the suggestion. The text has been amended accordingly.

  1. Thanks to your kind suggestion we have significantly restructured the case history section, dividing it into subchapters as suggested by the reviewer. We hope these corrections will facilitate the reading and understanding of our article.

  1. We did not move the literature review part as suggested as we believe that the aim of this section includes putting our case in the context of the knowledge available from other literature.

  1. We rephrased “The patient was moved to the Cardiological Rehabilitation Department right after discharge.” into “The patient was transferred or was addressed”, as you suggested

  1. A thorough English editing was performed by a native English speaker. He read and revised all our manuscript, and we incorporated all his amendments to the new draft of the manuscript.

  1. Unfortunately, no pertinent imaging from our own case report is available anymore due to a renewal of the informatic system in the Hospital where our patient was operated.

We truly hope you will find this new version of the manuscript to be more organized and understandable.

Thank you for your help, we hope that you will read and appreciate our final paper.

Reviewer 3 Report

In the present manuscript, Mule et al. review literature on T. asahii endocarditis. They also present in detail a relative case report. The manuscript is well written and covers all aspects.

I have one observation:

The method used for T. asahii identification is missing. I would recommend that this information is added. Also, a comment on methods (conventional, MALDI-TOF) used for Trichosporon identification would be helpful.

 Minor remarks

Line 290-294: "In our case, amphotericin B was chosen for induction therapy and obtained a good outcome, as in other case reports[19,23]. Also, combination therapy with 5-fluorocytosine and amphotericin B has also been associated with good clinical response in one case report[29]. Nevertheless, in other studies amphotericin B has been correlated to poor clinical outcome, and therefore it may not be a good option as a monotherapy[40,41]."

 instead of

"In our case, amphotericin B was chosen for induction therapy and obtained a good outcome, as in other  case reports[19,23]. On the other hand, amphotericin B has been correlated to poor clinical  outcome, and therefore it may not be a good option as a monotherapy[40,41]. For example, combination therapy with 5-fluorocystine and amphotericin B has also been associated  with good clinical response in one case report[29]."

  Line 294: "5-flucytosine" or "5-fluorocytosine" instead of "5-fluorocystine"

Author Response

Dear reviewer,

I would like to thank you very much for your kind revision on behalf of all the authors. We appreciate your contribution and your positive evaluation of our work.

I hope to fully answer your questions as follows:

  • T- asahii was identified through a conventional method. Thanks to your observation we will add this information to our paper.
  • We modified lines 290-294 as you suggested.

Thank you for your help, we hope that you will read and appreciate our final manuscript.

Round 2

Reviewer 2 Report

Dear authors,

Congratulations!

The manuscript has improved a lot after the revision.  All of my comments have been addressed properly.

Therefore, I agree with publication in the present form.

Kind regards, 

Minor corrections only